# Super-Ballistic Width Dependence of Thermal Conductivity in Graphite Nanoribbons and Microribbons

**DOI:** 10.3390/nano13121854

**Published:** 2023-06-13

**Authors:** Xin Huang, Satoru Masubuchi, Kenji Watanabe, Takashi Taniguchi, Tomoki Machida, Masahiro Nomura

**Affiliations:** 1Institute of Industrial Science, The University of Tokyo, Tokyo 153-8505, Japan; 2Research Center for Electronic and Optical Materials, National Institute for Materials Science, 1-1 Namiki, Tsukuba 305-0044, Japan; 3Research Center for Materials Nanoarchitectonics, National Institute for Materials Science, 1-1 Namiki, Tsukuba 305-0044, Japan

**Keywords:** phonon hydrodynamics, thermal conductivity, width dependence, graphite, nanoribbons, microribbons

## Abstract

The super-ballistic temperature dependence of thermal conductivity, facilitated by collective phonons, has been widely studied. It has been claimed to be unambiguous evidence for hydrodynamic phonon transport in solids. Alternatively, hydrodynamic thermal conduction is predicted to be as strongly dependent on the width of the structure as is fluid flow, while its direct demonstration remains an unexplored challenge. In this work, we experimentally measured thermal conductivity in several graphite ribbon structures with different widths, from 300 nm to 1.2 µm, and studied its width dependence in a wide temperature range of 10–300 K. We observed enhanced width dependence of the thermal conductivity in the hydrodynamic window of 75 K compared to that in the ballistic limit, which provides indispensable evidence for phonon hydrodynamic transport from the perspective of peculiar width dependence. This will help to find the missing piece to complete the puzzle of phonon hydrodynamics, and guide future attempts at efficient heat dissipation in advanced electronic devices.

## 1. Introduction

The effective manipulation of thermal phonons in nanoscale and microscale electronic devices has been of ongoing interest to researchers in the semiconductor industry [1,2]. Nowadays, with the increasing attention being given to environmental issues and the demand for energy resources, thermoelectrics has been developed as a promising energy harvesting technology involving direct conversion of thermal energy into electrical energy. For example, nanostructured thermoelectric generators (TEGs) based on Si materials have been designed for simple and economical power generation [3]. By matching or mismatching of the vibration spectrum of the terminal interfaces, to conduct or block the heat flow, a stable “on” or “off” state can be formed, thus making thermal logic operation possible for computing [4]. In addition, the development of lithography technique with a thermally controlled scanning tip has demonstrated its potential for high-resolution and large-throughput nanostructure patterning on the substrates [5].

In addition, another important objective of thermal management is the efficient dissipation of unexpected heat, to prevent device overheating. In recent years, the ballistic behavior of phonons has been exploited to achieve this goal, and many investigations have been carried out on different structures and materials [6,7,8,9,10,11]. With rare internal scattering between other partners, ballistic phonons can travel for a relatively long distance (up to 15 µm [12]), yet leave no temperature trace in the structures during their propagation.

However, a ballistic property requires the dominance of long-wavelength phonon modes, which can only be populated at extremely low temperatures, as their mean free path is greatly reduced at higher temperatures, due to the frequent Umklapp or defect scattering. On a parallel track, rapid heat dissipation in solid-state materials driven by the hydrodynamic properties of phonons has been frequently revisited and gradually realized by researchers, as another promising candidate for modern thermal management at elevated temperatures [13,14,15,16,17,18,19,20].

Phonon hydrodynamics, analogous to fluid dynamics (or hydrodynamics), is a field of research that focuses on the extraordinary collective behavior of momentum-conserved phonons in solids. The phonon Poiseuille flow [16,20] and the second sound [13,17,18] in steady and transient regimes are the two representative phenomena. Normal phonon–phonon scattering, which preserves the momentum-conserved nature, is an indispensable condition for demonstrating phonon hydrodynamic conduction. Recent studies have shown that normal scattering in graphite material is sufficiently large, due to its high Debye temperature, whereas Umklapp scattering is absent in small wavevector states and in the strong anharmonic interactions of the interlayers [21,22]. It is recognized that graphite possesses even more substantial hydrodynamic properties than its two-dimensional counterpart (i.e., graphene), owing to the additional channels for stronger three-phonon normal scattering participated by out-of-plane modes (i.e., bending acoustic modes (BA)) in the hydrodynamic temperature window [22].

Thermal transport modeling in a hydrodynamic regime was first described by the macroscopic hydrodynamic equations derived from the phonon Boltzmann transport equation (BTE) in common bulk crystals [23]. While the accurate prediction of thermal transport in nanostructures and microstructures requires a direct solution of the phonon BTE, the phonon collision term is very complex, making the solution of the phonon BTE very difficult. To simplify the problem, single-mode relaxation time approximation (SMA) has been used, to solve the phonon BTE in most cases where phonons are treated to be independent of each other during the scattering process [24,25]. Previous studies have shown that the SMA is a reasonable simplified model for obtaining thermal conductivity when the normal process is weak or negligible in common crystals [26,27]. A peculiarity is that what lies behind hydrodynamic phonon transport is the dominance of momentum-conserved collective phonon transport over individual phonon transport, which undergoes strong momentum-unconserved resistive scattering; therefore, Callaway’s dual relaxation approximation allows the correct description of phonon transport in systems with significant normal processes, such as graphite and graphene [13,28], and has been used extensively to model hydrodynamic thermal transport, by both semi-analytical [22] and numerical [28] methods.

It is obvious that hydrodynamic phonon transport (or the dominance of momentum-conserved normal scattering) is highly dependent on the temperature, and this dependence in graphite has recently been studied extensively [20,22,28,29,30]. Moreover, similar to the viscous effect in fluid dynamics, in an ideal hydrodynamic phonon flow, the diffuse scattering of phonons at the boundaries is the only resistive process: thus, the boundary of the structure plays a crucial role in hydrodynamic transport. Theoretical work has predicted a peculiar width dependence of the thermal conductivity in graphene [31] and graphite [22], separately, and has explained it by the phonon viscous damping effect in the phonon hydrodynamic regime; however, direct observation of the width dependence of thermal conductivity in graphite or graphene remains experimentally challenging.

For this paper, we carried out a pioneering experimental study of thermal conduction in free-standing graphite ribbon structures fabricated from the same graphite flake, with widths ranging from nanoscale to microscale. Based on a µs time-domain thermoreflectance (µ-TDTR) technique, we studied the width dependence of thermal conductivity over a wide temperature range of 10–300 K, to explore the different thermal conduction characteristics and their dependence on the structure width in ballistic, hydrodynamic and diffusive regimes. We aimed to provide an experimental support and a deeper insight into the width dependence of phonon hydrodynamics, and future directions in which to exploit the peculiar physics of collective phonons in thermal applications.

## 2. Materials and Methods

In addition to ${}^{12}$C, which occupies ∼98.9% of naturally occurring isotopes in carbon materials, ${}^{13}$C, as a secondary stable carbon isotope, accounts for the remaining ∼1.1%: its effect in suppressing thermal transport, due to sufficient momentum-unconserved isotope–phonon scattering, has been intensively claimed in regard to graphene [32], graphite [20], and diamond [33]. In order to avoid the additional isotopic effect on our investigation of hydrodynamic thermal transport, we used a world-class graphite crystal with an enriched ${}^{12}$C content of 99.98% in the current work.

The raman spectroscopy experiment measured that the thermal conductivity of free-standing graphene exceeds 2500 W·m${}^{-1}$·K${}^{-1}$ [34] at room temperature, while the thermal conductivity of graphene supported by copper is dramatically reduced to 370 W·m${}^{-1}$·K${}^{-1}$. The presence of a supporting substrate results in significant phonon–substrate interactions, which tend to affect the study of intrinsic thermal properties in the sample: hence, in order to avoid the considerable heat loss to the supported substrate [35], and to ensure that the heat was only dissipated through the structures of interest, the fabrication of suspended graphite ribbons was indispensable for the precise investigation of phonon hydrodynamics in this work.

To this end, we first mechanically exfoliated 150 nm thick flakes from the mother graphite flake, and bonded them to a 2.5-µm-thick SiO${}_{2}$ sacrificial layer on an Si substrate. In order to study the steady-state hydrodynamic heat transport in the graphite ribbons, the desired ribbon length is expected to be at least 5 times larger than the ribbon width, to ensure the formation of phonon hydrodynamic behavior and the observation of the hydrodynamic flow of phonons, as predicted by a recent work [28]: thus, the typical size of a flake required to pattern the ribbon structures is ∼100 × 100 µm${}^{2}$. A symmetrical airbridge configuration was designed, with a ribbon length of 30 µm, connecting the transducer and the heat sinks from both sides, as shown in Figure 1: this allowed us to obtain the thermal properties in the in-plane direction, which was of interest to the current work, and which allowed the heated structure to be sufficiently cooled down before the next heating pulse, for accurate measurement. The thermal properties of many other organic [36] and inorganic [11] nanomaterials and micromaterials have been investigated using a similar airbridge method.

To demonstrate the width dependence of thermal conductivity, and to maintain the precision of the widths in this work, we used an JEOL JBX-6300FS electron-beam lithography (EBL) system, with an overlay accuracy of 9 nm or less, to pattern the designed structures on the graphite flake. A 6 × 6 µm${}^{2}$ island was patterned in the center of the ribbons on the same flake, to load the transducer for thermal measurement in the following process. After the EBL patterning of the structures, we combined an electron beam physical vapor deposition (EBPVD) and a standard metal lift-off process, to deposit and form 100-nm-thick Al airbridge structures as transfer masks. The Al ribbon patterns were then transferred to the underlying graphite flake, using a reactive ion etching (RIE) method to etch the remaining graphite by O${}_{2}$ plasma. After the RIE process, we applied the Al etchant with mixed acids, to remove the Al masks and release the desired graphite ribbon structures on the SiO${}_{2}$ substrate.

Unlike the suspension process of samples fabricated on a semiconductor-on-insulator wafer—where vapor-phase hydrofluoric (VHF) acid enters through slits opened on the semiconductor device layer, and removes specific parts of SiO${}_{2}$ underneath, to suspend the structure (as demonstrated in previous works [37,38])—the suspension of graphite ribbon structures exfoliated and transferred onto SiO${}_{2}$ substrate is more challenging and complex. Metal VHF stoppers are required, to protect the graphite ribbons from falling down to the substrate or peeling off during the SiO${}_{2}$ removal, because the etching rate of SiO${}_{2}$ under the metal is lower than that under the graphite. To this end, we combined the laser lithography and EBPVD, to fabricate two 250 × 400 µm${}^{2}$ Au heat sinks, which were used as metal VHF stoppers, to cover the graphite ribbons from both sides. A 70-nm-thick round Al pad was then deposited at the center of the graphite bridge by another EBPVD process, as a transducer for optical/thermal signal detection. Finally, we applied VHF acid, to remove the SiO${}_{2}$ layer, in order to suspend the patterned ribbon structures on the substrate.

## 3. Results

To study the thermal conduction in the graphite ribbons, we used the pump–probe method, in order to characterize the thermal properties of our graphite ribbons in a TDTR setup on the microsecond scale: as a non-contact measurement technique, this significantly minimized the difficulties of sample preparation (no electrical heaters and sensors), and allowed rapid measurement on a real-time scale. This approach has been widely used to study phonon transport, and to evaluate the thermal properties in various graphitic materials with high thermal conductivity, such as graphite [17,18,39] and diamond [40,41].

During the measurement, our samples were placed in a cryostat with a high vacuum level (<10${}^{-5}$ Pa), to avoid unexpected heat loss to the ambient environment. Inside, a metal heating plate and a liquid helium flow system were used, to ensure the precise adjustment of the sample temperature. We first induced excitation in the Al transducer, using a pump laser (λ = 642 nm) with a 10 µs repetitive pulse at a rate of 1 kHz. Meanwhile, a probe laser (λ = 785 nm) continuously detected the optical response of the transducer. The thermoreflectance measurements were based on the temperature dependence of the material’s reflectivity, where the reflectivity of the metal transducer followed the same responses as the temperature of the transducer [42], as Δ*R*/*R* = $C_{th}$Δ*T*, where $C_{th}$ was the thermoreflectance coefficient. During the pump laser heating, the temperature-dependent reflectivity changed dramatically, as shown by the increase in Δ*R*/*R* (the normalized thermoreflectance signal) in Figure 2a.

The conductive Al transducer and the underlying graphite structure reached thermal equilibrium in less than 1 µs, due to the low thermal boundary resistance, as investigated in our previous work [20]: this allowed the heat to be conducted through the graphite structures on a much longer timescale, where the phonon contribution to the thermal conductivity was dominant. In semiconductors, such as the graphene and graphite mentioned in the current work, or insulators (e.g., organic materials), electrons carry a relatively small or even negligible amount of the heat [30,43,44], as determined by the Wiedemann–Franz law ($\kappa_{e}$ = *L*σ*T*, where *L* is the Lorenz number), using the experimentally measured electron conductivity (σ).

After removal of the pump pulse, the reflectivity change shows exponential decay (exp(−t/τ)), as heat is dissipated through the graphite ribbon structures, where the decay time τ is a crucial parameter for characterizing the thermal property. At each temperature point, we performed 3–4 measurements for each ribbon structure, to collect sufficient decay time data to extract the thermal conductivity, as indicated by the error bars of the thermal conductivity results in the figures.

We then carried out finite element method modeling, to simulate the heat dissipation through the structures, and to reproduce the decay times measured in our experiments. To extract the thermal conductivity of our samples, we swept a range of thermal conductivities, and their corresponding decay times were also obtained, by fitting the exponential decay curves from the simulations. The in-plane thermal conductivity values of the measured graphite ribbons were thus extracted, by interpolating a linear function between the simulated thermal conductivity and the measured decay time. A detailed method for extracting the thermal conductivity can be found in our previous works [37,38].

To investigate the width dependence of the thermal transport in our graphite samples, we fabricated and measured the thermal conductivity of graphite ribbons with different widths of 300 nm, 600 nm, and 1.2 µm, as shown in Figure 2b. We used an ultra-high-resolution scanning electron microscope (SEM) (Hitachi SU9000) to characterize the dimensions of our structures. Although it was not possible to accurately measure the roughness by SEM, our high-resolution images show that the roughness of the edges was no more than 50 nm, as detailed in Appendix A.

Due to the low atomic mass of carbon, and the strong intraplanar sp${}^{2}$ bonding of carbon atoms, heat in graphite conducts more efficiently in the direction parallel to the basal plane: this results in much higher thermal conductivity than that of other common three-dimensional solid-state materials. Experimentally measured in-plane thermal conductivity of bulk-like graphite ranges from ∼1370 W·m${}^{-1}$·K${}^{-1}$ to ∼1950 W·m${}^{-1}$·K${}^{-1}$ at room temperature [30,45,46]. As shown in Figure 2c, our measured value of thermal conductivity in a 1.2-µm-wide graphite ribbon was 1111 W·m${}^{-1}$·K${}^{-1}$. As the ribbon width was further reduced, the thermal conductivity followed a continuous decreasing trend, dropping to 733 W·m${}^{-1}$·K${}^{-1}$ in a 300-nm-wide graphite ribbon, where the diffuse boundary scattering of phonons became stronger, due to the reduction of the structure size. The strong dependence of the thermal conductivity on the characteristic size (i.e., width) of the pure graphite sample was also well-demonstrated by Fugallo et al., in their calculations with the exact solution of the Boltzmann transport equation for phonons [25]. The rough edges of the structures may also explain the lower experimentally measured thermal conductivities in this work, compared to the calculated results; however, a preferable agreement was still found, in terms of the qualitative trend of the width dependence.

By contrast, heat conduction in the direction perpendicular to the basal planes was largely limited by the weak interplanar van der Waals coupling between the adjacent graphene layers. The thermal conductivity along the out-of-plane direction of pyrolytic graphite is ∼6 W·m${}^{-1}$·K${}^{-1}$ at room temperature [45], which is ∼300 times lower than that along the in-plane direction. Meanwhile, previous works have observed a transition of thermal conductivity from graphene to graphite within a few layers of increasing graphene thickness [47,48]. The thickness of our graphite ribbon was 150 nm (∼450 graphene monolayers): thus, we assumed a negligible effect of the multilayer graphite on the in-plane thermal transport, due to the weak van der Waals interaction along the out-of-plane direction.

As shown above in Figure 2c, the thermal conductivity decreased monotonically with the narrowing of the ribbon width at 300 K; however, a previous work predicted that the width dependence of thermal conductivity varies according to temperature in different heat transport regimes of graphite [22]. In contrast to diffusive heat transport at higher temperatures, the phonon flux is rapidly damped by momentum-unconserved Umklapp processes. At lower temperatures, with the absence (or negligibility) of Umklapp scattering, it enters the hydrodynamic regime, where the thermal phonons tend to conserve their momenta, and behave as a whole, in a collective motion, under the frequent momentum-conserved normal processes. At sufficiently low temperatures, the phonons are mainly populated in the long wavelength states. Individual phonons rarely experience internal scattering, but are largely limited by the boundaries of the structures.

To this end, we performed temperature-dependent thermal conductivity measurements of graphite ribbons from 300 K down to 10 K, using the µ-TDTR setup, which covered the above-mentioned phonon transport regimes [22,30,39]. As shown in Figure 3, with the temperature drop from 300 K, the Umklapp phonon–phonon scattering was gradually hindered, resulting in an increase in thermal conductivity in the 1.2-µm-wide graphite sample. However, the thermal conductivity enhancement was less pronounced in the 300-nm-wide and 600-nm-wide cases, indicating that diffuse boundary scattering is comparatively strong in such narrow structures, even at high temperatures. A similar invariance of the thermal conductivity with decreasing temperature has also been observed in Si thin film with nanocone structures, due to the sufficient boundary scattering of phonons [37]. At temperatures below 150 K, the thermal conductivities of the three samples generally followed the same trend, with temperature further decreasing; however, a significant difference in the thermal conductivity of the three samples was shown throughout the temperature range. It is noteworthy that a stronger width dependence of the thermal conductivity was observed in the intermediate temperature range (i.e., ∼30–80 K) in the log–log plot.

Many recent works have demonstrated the different temperature-dependent behavior of thermal conductivity compared to the ballistic limit in hydrodynamic phonon transport regimes in graphite [20,22,29]. On the other hand, another important aspect in proving hydrodynamic phonon flow is the super-ballistic width dependence of thermal conductivity [16,31], as indicated by the effective mean free path ($\Lambda_{eff}$ ∼ $W^{2}/\Lambda_{N}$), from the random walk theory [49,50], where *W* and $\Lambda_{N}$ are the sample width and the phonon mean free path of the normal process, respectively. Thus, in a hydrodynamic regime, thermal conductivity (κ ∼ $\Lambda_{eff}$) is proportional to $W^{\alpha }$ (α > 1), due to significant normal scattering ($\Lambda_{N}<W$). By contrast, in a ballistic regime, thermal conductivity is linearly related to the width of the structure, so that κ ∼ $W^{\alpha }$ (α=1). Therefore, the superlinear width dependence of thermal conductivity is also considered to be robust evidence for the phonon Poiseuille flow.

As shown in Figure 4a, we plotted our experimentally measured thermal conductivity as a function of graphite ribbon width at several typical temperatures covering the entire phonon transport regimes of ballistic, hydrodynamic and diffusive. In principle, the thermal conductivity solely increases with the widening of the graphite ribbon structure over the entire temperature range; however, the rate of increase (or slope) varies for different thermal transport regimes. Here, to illustrate the width dependence, we fitted the data points with linear functions at each temperature point, as shown by the solid lines. Thermal conductivity was thus related to the width of the graphite ribbon, using a simple function: κ ∼ $W^{\alpha }$; therefore, we defined the exponent α as the coefficient that indicated how strongly the thermal conductivity depended on the width at different temperatures.

As shown in Figure 4b, we compared the values of the width dependence coefficient (α) at different temperatures in the temperature range from 10 to 300 K. At 10 K, thermal conduction was typically in the ballistic regime, where ballistic phonons are mainly restricted by the size of the structures, and thermal conductivity is linearly proportional to width (i.e., κ ∼ *W*); therefore, α is ideally equal to 1. However, in the actual experiment, due to other unavoidable defect phonon scattering mechanisms, we obtained a coefficient of ∼0.54 as a base value for the ballistic case.

As the temperature increased, the scattering processes between the phonons were gradually excited. When the momentum-conserved normal phonon–phonon processes were sufficient, it led to the peculiar collective transport of phonons in the hydrodynamic regime. At higher temperatures, normal scattering became more significant, the phonon mean free path was further reduced, and the thermal conductivity showed stronger width dependence, compared to that in the ballistic transport regime (i.e., κ ∼ $W^{2}/\Lambda_{N}$), as shown by the increase of α following the elevation of the temperature in Figure 4b. At 75 K, the Umklapp scattering was still neglectable, the normal scattering dominated over other momentum-unconserved scatterings, and we observed the strongest width dependence, indicating the prominent hydrodynamic behavior of the phonons. After reaching its peak, the coefficient α dramatically decreased with further increase in temperature, indicating a smaller width dependence when Umklapp scattering was predominant, and hydrodynamic flow was ultimately destroyed in the diffusive regime at high temperatures.

## 4. Discussion

Fluid dynamics deals with the motion of the fluid as a continuum under force, and the interaction between the fluid and the boundaries [51]. A commonly discussed case in modern hydrodynamics is the steady-state flow of an incompressible fluid in a cylindrical tube, where the transfer rate of the molecular Poiseuille flow varies as the fourth power of the radius of the tube [52]. Similarly, investigation of the transfer rate of the phonon Poiseuille flow has established that it follows the same power law of the radius as that of the molecular flow in the circular tube in previous works [16,23]. Therefore, for a phonon Poiseuille flow in a three-dimensional rectangular ribbon structure, as was the case in the present work, the rate of heat flow scales as the cube of the width. Correspondingly, the thermal conductivity should follow the square of the width due to the hydrodynamic flow of phonons; however, the $W^{2}$ trend could only occur by considering no Umklapp scattering, such as that in fluid dynamics. In a recent work, a width dependence of $W^{1.17}$ was obtained at 100 K in a suspended graphene ribbon, using the Monte Carlo method [31], while it dropped to $W^{0.17}$ at 300 K, due to the Umklapp scattering being stronger than the hydrodynamic effect at sufficiently high temperatures. Therefore, a superlinear width dependence in thermal conductivity clearly indicates hydrodynamic phonon transport.

However, although we did not observe the superlinear (α > 1) width dependence of the thermal conductivity, due to the additional momentum-unconserved scattering of phonons that was induced during the sample preparation, we could see that the width-dependent coefficient α within the hydrodynamic temperature window (i.e., 45–75 K) was significantly more pronounced than that outside the window, both in the ballistic (10 K) and diffusive (300 K) regimes, as shown in Figure 4b. The width dependence of the thermal conductivity here showed an enhanced thermal conduction compared to that in the ballistic transport regime: this demonstrated the emergence of the hydrodynamic behavior of phonons in our carbon isotope-enriched graphite nanoribbons and microribbons. Moreover, superlinear width dependence represents an even stronger hydrodynamic phonon flow, and is more difficult to observe, due to the additional size effect from the length of the structure, as recently realized in a theoretical study of phonon hydrodynamic transport [28]. Further experimental attempts that focus on the ultimate carbon isotope purification and prolongation of the ribbon structure may be beneficial for the optimization of the phonon hydrodynamic transport, and for the observation of superlinear width dependence in graphite.

In this work, to explore the well-predicted prominent width dependence of hydrodynamic phonons, and to further confirm the existence of hydrodynamic thermal transport, we fabricated free-supporting nanoscale and microscale ribbons on isotopically enriched graphite flake. Super-ballistic width dependence of the thermal conductivity was observed in our experimental results at 75 K, showing that potential hydrodynamic behavior may occur. This provides an adequate complement to a recent work on the observation of phonon hydrodynamic flow [20] from a different point of view, by considering the width dependence of thermal conductivity, and a new direction in which to make phonon hydrodynamics practical in nanoscale and microscale heat management. Further experiments are needed, to investigate the extraordinary superlinear width dependence of thermal conduction in the hydrodynamic regime, based on the suspended microstructure system established in this work.

## Figures and Tables

**Figure 1 nanomaterials-13-01854-f001:**
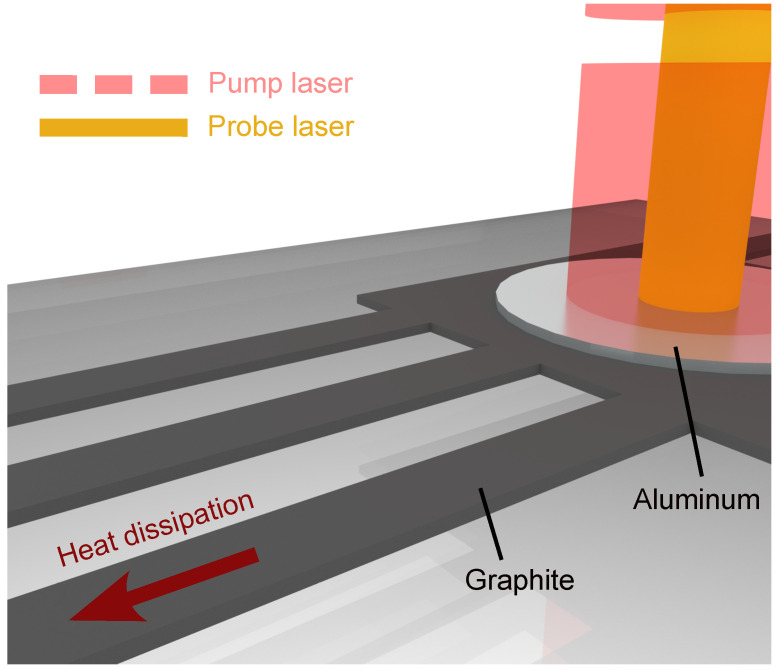
Thermal characterization of graphite nanoribbons and microribbons, using the pump–probe method.

**Figure 2 nanomaterials-13-01854-f002:**
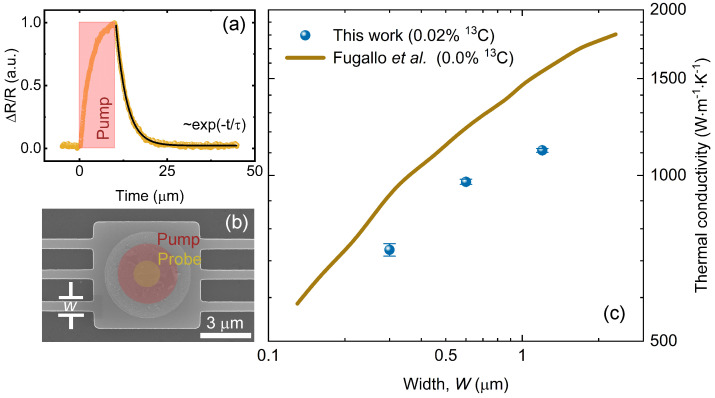
(**a**) Normalized thermoreflectance signal; (**b**) Scanning electron microscope (SEM) image of a suspended graphite ribbon with a width of 600 nm; (**c**) Experimentally measured thermal conductivity (dots) compared to calculated results from Ref. [25] (line), as a function of the graphite ribbon width at 300 K.

**Figure 3 nanomaterials-13-01854-f003:**
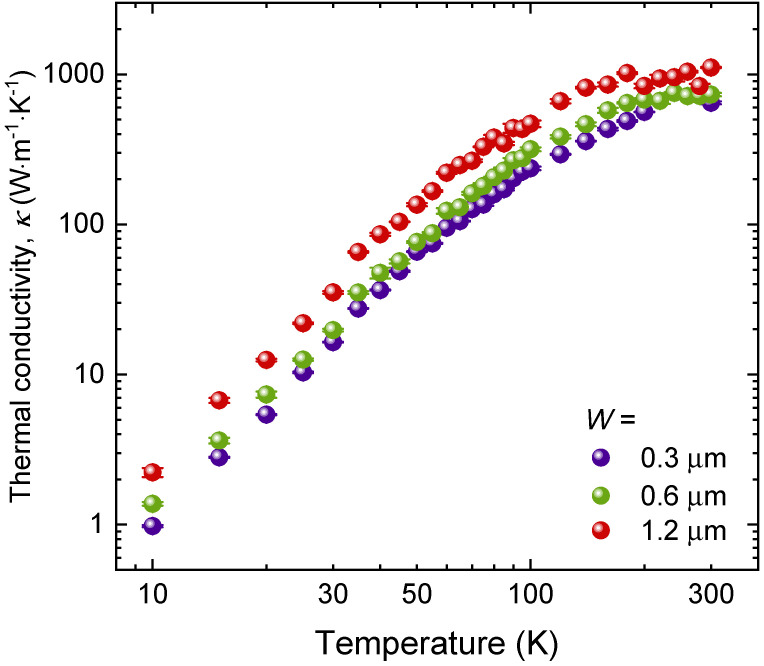
Thermal conductivity as a function of temperature for graphite ribbons of different widths (*W*).

**Figure 4 nanomaterials-13-01854-f004:**
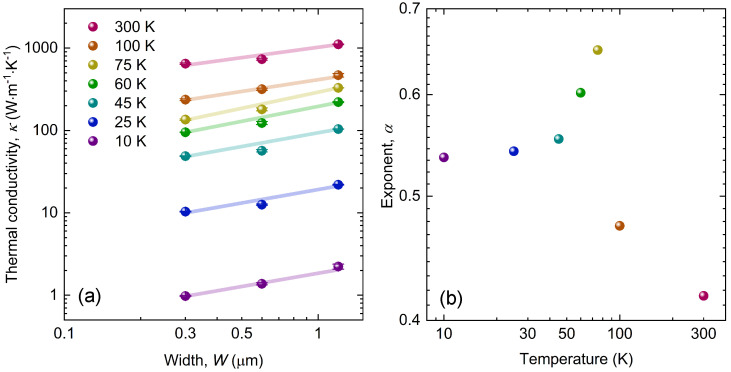
(**a**) Thermal conductivity as a function of graphite ribbon width at different temperatures; the lines show the linear fit to the measured data; (**b**) Coefficient of the width dependence, α, as a function of temperature.

## Data Availability

The data that support the findings of this study are available from the corresponding authors upon reasonable request.

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
