# Peer review of "Super-Ballistic Width Dependence of Thermal Conductivity in Graphite Nanoribbons and Microribbons"

_nanomaterials, 2023, doi:10.3390/nano13121854_

Round 1

Reviewer 1 Report

Very good article. Thanks.

I would just suggest that the authors change the name of the suspended structures, which in my opinion are micro and nanoribbons.

Author Response

Please see the attachment for a point-by-point response. 

Reviewer 2 Report

The authors present a very interesting and well described study of the size dependence of thermal conductivity for graphite nanowires in the regime where hydrodynamic phonon flow is expected. I think that this work is suitable for publication but their are some issues that the authors may address first. These are as follows.

It would be desirable to better describe the measurement and characterization of the width and the precision of its determination.

It would be desirable to present any available information on the roughness of the edges and discuss this.

There are some minor language issues that should be corrected, for example "importatn" on line 18 and "unexpected heat" when they probably mean unwanted heat.

The quality of the English is very good in this manuscript, but there are some minor issues as noted above. The authors may want to proof read the work before the final submission.

Author Response

(The authors gave the same response as above.)

Reviewer 3 Report

Authors have shown results pertaining to phonon hydrodynamic transports. Regarding the fabricated structure shown in figure 1, symmetrical design and fabrication of graphite wires was performed, thermal transport structure design is relevant to measure the decay time as shown in figure 2, would the authors elaborate as to the designed structure being optimized to measure or collect sufficient decay time data to calculate the thermal conductivity properties. 

Author Response

(The authors gave the same response as above.)

Reviewer 4 Report

This study focuses on the temperature dependence of thermal conductivity in solids and its relation to hydrodynamic phonon transport. The researchers experimentally measured thermal conductivity in graphite wire structures of varying widths and observed a stronger width dependence in the hydrodynamic window compared to the ballistic limit. This finding provides essential evidence for phonon hydrodynamic transport and contributes to understanding heat dissipation in advanced electronic devices.  The work can be publishable after addressing the following minor comments:

  1. There are a few thermal transport models, which are common in the community of thermal conductance research. Thermal transport modeling over the results reported in this work will strengthen the quality of this work.
  2. While major heat carrier in organics would be phonon, that of conductive electrode would be electron, I think. I wonder how to describe the major heat carrier in graphite. More discussion with related reference citations on the issue will improve the work.
  3. The following works involving thermal transport on organic nano-materials  are related to this work, and thus need to be cited: Nano Lett. 2010, 10, 5, 1645–1651;  J. Mater. Chem. A, 2020, 8, 19746-19767; Journal of Applied Physics 130, 070903 (2021); 2020 Nanotechnology 31 324003
  4. I wonder if there is any effect of multilayer graphite on the thermal transport.

Author Response

(The authors gave the same response as above.)

Round 2

Reviewer 4 Report

the issues have been addressed and the current draft can be publishable.